# X-Cardia: Phenotype-Guided Cross-Modal Alignment for Opportunistic Cardiac Screening on Routine Chest CT

**Nusrat Binta Nizam**[1]                                        NN284@CORNELL.EDU
**Fengbei Liu**[1]                                               FL453@CORNELL.EDU
**Sunwoo Kwak**[1]                                               SK3355@CORNELL.EDU
**Ilan Richter**[2]                                    IR2498@CUMC.COLUMBIA.EDU
**Jayant K Raikhelkar**[2]                            JKR2146@CUMC.COLUMBIA.EDU
**Ashley Beecy**[3]                                          ASHLEYBEECY@GMAIL.COM
**Nir Uriel**[2]                                       NU2126@CUMC.COLUMBIA.EDU
**Deborah Estrin**[1]                                          DESTRIN@CORNELL.EDU
**Mert R Sabuncu**[1,4]                                       MSABUNCU@CORNELL.EDU

[1] *Cornell Tech, New York, USA*

[2] *Columbia University Irving Medical Center, New York, USA*

[3] *Sutter Health, California, USA*

[4] *Weill Cornell Medicine, New York, USA*

**Editors:** Accepted for publication at MIDL 2026

## Abstract

Multimodal medical data offer an opportunity to learn general-purpose representations for cardiovascular diagnosis. We introduce X-Cardia, a cardiac phenotype-guided multimodal framework that uses structured data as intermediate supervision during pre-training. X-Cardia learns to extract cardiac information from non-contrast, non-gated chest CT scans by aligning CT features with tabular measurements derived from echocardiography (ECHO) and electrocardiography (ECG). Our method combines CLIP-style contrastive pre-training with a non-parametric Nadaraya–Watson (NW) prediction head that enforces phenotype-level similarity via exemplar-based alignment. Pre-training on 20,574 patients, followed by fine-tuning on ten cardiac abnormality prediction tasks, yields substantial performance gains. X-Cardia improves AUROC by up to 8% on the held-out test set and delivers an average 11.8% AUROC improvement in a 5-shot regime. These results demonstrate that explicit phenotype alignment produces interpretable, data-efficient representations and enables routine chest CT to support opportunistic cardiac screening. Code is available at: https://github.com/sumona00/X-Cardia.

**Keywords:** Multimodal Alignment, Phenotype-Guided, Nadaraya–Watson head, Chest CT, Echocardiography, Electrocardiography.

## 1. Introduction

Modern cardiac diagnostics routinely use volumetric scans such as CT, together with structured clinical measurements (e.g., chamber dimensions, pressure estimates) derived from different modalities, such as echocardiography and electrocardiography (Ota et al., 2001; Jenkins et al., 2009). These modalities provide complementary information: imaging captures morphological patterns, while tabular measurements encode expert-derived phenotypes (Henry et al., 1980; Lau et al., 2023; Ghorbani et al., 2020; Castrejon et al., 2016).

Despite this, current deep learning pipelines typically operate on a single modality or use simple late fusion, leaving the rich semantic correspondence between modalities largely untapped. This limits robustness and underutilizes valuable clinical structure. Routine non-gated, non-contrast chest CT scans represent an especially challenging but impactful setting for opportunistic cardiac screening. These studies are acquired for non-cardiac indications, lack cardiac gating, and attenuate cardiovascular detail—making cardiac interpretation difficult even for experts. Yet many clinically meaningful phenotypes, such as chamber dilation or valvular dysfunction, manifest jointly as geometric patterns on CT and quantitative deviations in structured measurements. Leveraging these natural correspondences could enable models to extract cardiac information from routine thoracic imaging.

Cross-modal alignment (Wang et al., 2022; Jiang and Ye, 2023) provides a mechanism to learn such shared semantic structure. Aligning imaging and tabular representations allows gradients from cleaner, more structured phenotypes to regularize the image encoder, guiding it toward physiologically grounded features that generalize across tasks and modalities. Prior multimodal contrastive learning (MMCL) (Yuan et al., 2021; Radford et al., 2021) approaches have demonstrated the promise of contrastive alignment, but often produce global, mixed embeddings that offer limited interpretability and weak few-shot generalization. Moreover, they rarely enforce per-phenotype similarity, which is crucial in cardiology where findings are sparse and modalities may disagree.

These challenges are amplified when the goal is to infer cardiac status from routine chest CT. Without explicit phenotype guidance, models may rely on shortcut features or collapse toward modality-specific biases, especially in data-scarce settings. Effective cardiac–non-cardiac transfer therefore requires an approach that (i) tightly couples CT and structured measurements, (ii) grounds representations in clinically meaningful phenotypes, and (iii) supports exemplar-based reasoning. In this work, we introduce X-Cardia, a cardiac phenotype–guided multimodal framework for aligning chest CT with ECHO and ECG-derived measurements. X-Cardia integrates a CLIP-style contrastive objective with a non-parametric Nadaraya–Watson (NW) head (Cai, 2001; Wang et al., 2023; Wang and Sabuncu, 2022) that enforces phenotype-level similarity via a support bank of exemplar embeddings. This hybrid objective produces interpretable, data-efficient representations that transfer effectively to downstream cardiac prediction tasks on non-gated chest CT.

We pre-train on a large cohort of 20,574 patients and fine-tune the CT encoder on ten clinically relevant abnormalities derived from ECHO. X-Cardia substantially improves performance over strong baselines—including standard multimodal contrastive learning, achieving up to 8% AUROC gains in full-data settings and nearly 11.8% improvements in 5-shot regimes. These results indicate that explicit phenotype alignment is key to unlocking cardiac information from routine CT and enabling scalable opportunistic screening.

## 2. Related Works

### 2.1. Learning with Tabular Data

Conventional tabular models such as XGBoost (Chen and Guestrin, 2016) and Light-GBM (Ke et al., 2017) remain strong baselines, often outperforming early neural networks (Shwartz-Ziv and Armon, 2022). Recent attention-based architectures better capture feature dependencies: TabTransformer (Huang et al., 2020) introduces contextual embed-

dings, and TabNet (Arik and Pfister, 2021) applies sequential attention for feature selection. Foundation models like TabPFN (Hollmann et al., 2022) and TabLLM (Hegselmann et al., 2023) enable few-shot reasoning through meta-learning or language-model serialization, producing transferable structured embeddings.

In medical applications, tabular representations offer complementary physiological information. Prior work has shown that contrastively coupling image and tabular encoders can enhance unimodal performance (Hager et al., 2023). Jiang et al. (Jiang et al., 2024a,b) extend this idea by aligning visual feature channels with clinical phenotypes using optimal transport and mutual information. These methods highlight the value of tabular data as a source of structured, expert-derived signals for guiding representation learning.

### 2.2. Cross-Modal Transfer

Cross-modal alignment learns a shared embedding space. For example, vision–language models such as CLIP (Radford et al., 2021), ALBEF (Li et al., 2021), IRRA (Jiang and Ye, 2023), CUSA (Huang et al., 2024), and UNITER (Chen et al., 2019) use contrastive or masked objectives to link image and text representations. Liang *et al.* (Liang et al., 2022) revealed a persistent modality gap formed by each encoder due to initialization and temperature dynamics.

In medical imaging, multimodal contrastive methods such as MMCL (Yuan et al., 2021; Hager et al., 2023), SimCLR-based approaches (Chen et al., 2020; Tang et al., 2020), and clinically grounded frameworks like CHARMS (Jiang et al., 2024a,b) demonstrate that structured signals can regularize visual encoders and improve data efficiency. The modality-focusing hypothesis (Xue et al., 2022) further suggests that cross-modal transfer succeeds when modalities share causal, modality-general features. Recent work on large vision–language models (Li et al., 2025) underscores the importance of aligning both latent spaces and model behavior.

Motivated by these, we perform cardiac-to-non-cardiac transfer by aligning chest CT with ECHO and ECG-derived phenotypes via a contrastive, phenotype-supervised objective, reducing modality gaps and yielding more data-efficient multimodal representations.

## 3. Methodology

We start with the basic setup in Sec. 3.1, followed by the encoders and representation fusion in Sec. 3.2. We then describe the Nadaraya–Watson head in Sec. 3.3, the support bank construction in Sec. 3.4, and the cross-modal alignment in Sec. 3.5. The training and evaluation details are described in Sec. 3.6. Finally, we present the supervised fine-tuning on cardiac binary targets in Sec. 3.7. An overview of the proposed cross-modal pre-training and fine-tuning framework is shown in Figure 1.

### 3.1. Problem Setup

We denote our multimodal training dataset as $\mathcal{D} = \{(\mathbf{x}_i^{\text{img}}, \mathbf{x}_i^{\text{tab}}, \mathbf{y}_i)\}_{i=1}^{|\mathcal{D}|}$, where $\mathbf{x}_i^{\text{img}} \in \mathcal{X} \subset \mathbb{R}^{D \times H \times W}$ is the 3D chest CT volume with $D$ as number of slices, $H$ and $W$ as height and width. $\mathbf{x}_i^{\text{tab}} \in \mathcal{X} \subset \mathbb{R}^F$ is the structured tabular feature vector derived from ECHO and

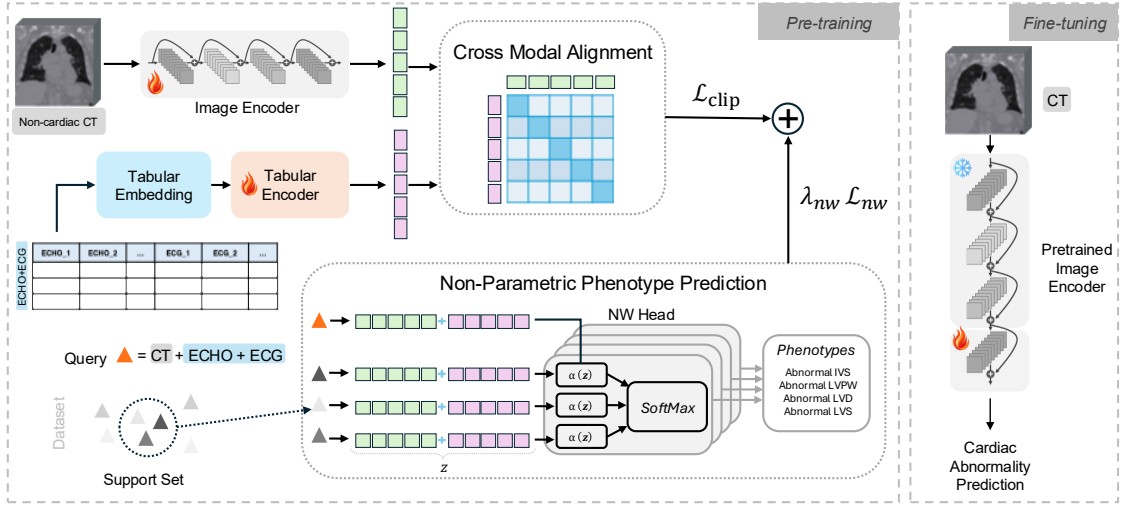

Figure 1: Overview of proposed X-Cardia. During pre-training (left), multimodal alignment is learned across chest CT and tabular features (ECHO and ECG) using a CLIP-style contrastive loss $\mathcal{L}_{\text{clip}}$ and cardiac supervision via a Nadaraya–Watson (NW) head with loss $\mathcal{L}_{\text{nw}}$. Flame icon means learnable, whereas snowflake icon means frozen. During fine-tuning (right), only the last layer of pre-trained CT encoder is optimized to predict cardiac abnormalities from non-cardiac chest CTs, leveraging the aligned and phenotype-guided latent space obtained during pre-training.

ECG with $F$ as the number of features. $\mathbf{y}_i \in \{0,1\}^C$ is the multi-hot vector and $C$ is the number of classes.

### 3.2. Encoders and Representation Fusion

**Image encoder.** We denote image encoder (3D ResNet–50 (He et al., 2016; Hara et al., 2017; Wang et al., 2018)) as $f_\theta : \to \mathbb{R}^d$ mapping $\mathbf{x}_i^{\text{img}}$ to a $d$-dimensional embedding:

$$\mathbf{e}_i^{\text{img}} = f_\theta(\mathbf{x}_i^{\text{img}}) \tag{1}$$

where $\theta$ represents the learnable parameters of the image encoder and $\mathbf{e}_i^{\text{img}} \in \mathbb{R}^d$ is the image embedding after Global Average Pooling (GAP) and MLP head.

**Tabular encoder.** We adopt a FT-Transformer (Gorishniy et al., 2021) based architecture. We define a tokenizer $h_\psi$ and encoder $g_\phi$ that map the input $\mathbf{x}_i^{\text{tab}} \in \mathbb{R}^F$ to an embedding $\mathbf{e}_i^{\text{tab}} \in \mathbb{R}^d$. The tokenizer $h_\psi$ projects each scalar feature to form a sequence of embeddings. The encoder $g_\phi$ processes this sequence via Transformer layers, aggregates the output via GAP, and applies a final projection head:

$$\mathbf{e}_i^{\text{tab}} = g_\phi\big(h_\psi(\mathbf{x}_i^{\text{tab}})\big), \tag{2}$$

where $\psi$ and $\phi$ are learnable parameters.

**Embedding fusion.** We use sum fusion for balanced optimization of both encoders and because it performs comparably to more complex strategies in the analysis presented in Appendix E. We further fuse $\mathbf{e}_i^{\text{img}}$ and $\mathbf{e}_i^{\text{tab}}$ to obtain a shared representation for phenotype prediction. We define the fusion operation as follows:

$$\mathbf{z}_i = \frac{\mathbf{e}_i^{\text{img}} \oplus \mathbf{e}_i^{\text{tab}}}{\left\| \mathbf{e}_i^{\text{img}} \oplus \mathbf{e}_i^{\text{tab}} \right\|_2} \tag{3}$$

where $\oplus$ denotes element-wise summation and $\|\cdot\|_2$ is $\ell_2$ norm.

### 3.3. Non-Parametric Phenotype Prediction (Nadaraya–Watson Head)

We adopt a Nadaraya–Watson (NW) estimator (Cai, 2001; Wang et al., 2023; Wang and Sabuncu, 2022) on fused embedding $\mathbf{z}_i$ for phenotype prediction. The NW head is non-parametric and has no learnable parameters; it uses support bank $\mathcal{D}_{sup} = \{(\hat{\mathbf{z}}_k, \hat{\mathbf{y}}_k)\}_{k=1}^{|\mathcal{D}_{sup}|}$, where $\hat{\mathbf{z}}_k$ is fused embedding and $\hat{\mathbf{y}}_k$ is the multi-hot phenotype label of sample $k$. Given a query embedding $\mathbf{z}_i$, we first compute temperature-scaled similarities,

$$\alpha_k(\mathbf{z}_i) = \frac{\exp\left((\mathbf{z}_i \cdot \hat{\mathbf{z}}_k)/\tau_{\text{nw}}\right)}{\sum_{j=1}^{|\mathcal{D}_{sup}|} \exp\left((\mathbf{z}_i \cdot \hat{\mathbf{z}}_j)/\tau_{\text{nw}}\right)}, \tag{4}$$

where $\tau_{\text{nw}}$ is a temperature hyperparameter controlling the sharpness of the attention distribution over support, $\alpha_k(\mathbf{z}_i)$ denotes the weight assigned to query embedding $\mathbf{z}_i$ over support bank. The predicted phenotype probabilities are then given by:

$$\hat{\mathbf{p}}_i = \sum_{k=1}^{|\mathcal{D}_{sup}|} \alpha_k(\mathbf{z}_i)\, \hat{\mathbf{y}}_k, \tag{5}$$

where $\hat{\mathbf{p}}_i$ is the $C$-dimensional vector of phenotype probabilities for query $\mathbf{z}_i$, which can be viewed as weighted average of the support-label vectors in phenotype space. For training the NW head, we use binary cross-entropy (BCE) loss $\mathcal{L}_{\text{nw}}$ applied to all cardiac phenotypes and averaged over the training set.

### 3.4. Support Bank Construction

We maintain a non-parametric support bank for Nadaraya–Watson inference on embeddings. At initialization, all training samples are encoded and up to $K$ *positive* examples per phenotype are selected: if a phenotype has fewer than $K$ positives, we keep all; otherwise, we run $k$-means with $K$ clusters and retain the samples closest to each centroid. The resulting normalized support embeddings are stored in the NW head and used to compute cosine similarities between query and support embeddings, and the bank is rebuilt every $M$ epochs to track the evolving representation.

### 3.5. Cross-Modal Alignment

To align image and tabular modalities in a shared embedding space, we employ a CLIP-style symmetric contrastive loss (Oord et al., 2018; Radford et al., 2021; Hager et al.,

2023). We define scaled similarities as $s_{ij} = (\mathbf{e}_i^{\text{img}} \cdot \mathbf{e}_j^{\text{tab}})/\tau_{\text{clip}}$ where, $\tau_{\text{clip}}$ is a temperature hyperparameter controlling sharpness of the similarity distribution. Cross-modal loss is,

$$\mathcal{L}_{\text{clip}} = -\frac{1}{2|\mathcal{D}|} \left[ \sum_{i=1}^{|\mathcal{D}|} \log \frac{\exp(s_{ii})}{\sum_{j=1}^{|\mathcal{D}|} \exp(s_{ij})} + \sum_{j=1}^{|\mathcal{D}|} \log \frac{\exp(s_{jj})}{\sum_{i=1}^{|\mathcal{D}|} \exp(s_{ji})} \right] \tag{6}$$

which encourages each matched image–tabular pair (the diagonal terms $s_{ii}$) to have higher similarity than all non-matching pairs within the batch.

### 3.6. Training and Evaluation

The overall objective combines cross-modal alignment and phenotype supervision as:

$$\mathcal{L}_{\text{total}} = \mathcal{L}_{\text{clip}} + \lambda_{\text{nw}}\, \mathcal{L}_{\text{nw}}, \tag{7}$$

The weighting term $\lambda_{\text{nw}}$ is tuned to balance alignment and phenotype learning. Training uses AdamW with cosine-annealed learning rate. Optimization runs for up to 100 epochs with early stopping criteria. In all the comparison studies, MMCL is actually our model (X-Cardia) without NW head and trained using only $\mathcal{L}_{\text{clip}}$ loss.

### 3.7. Supervised Fine-Tuning on Cardiac Binary Targets

We fine-tuned the pre-trained image encoder (3D ResNet-50 (He et al., 2016; Wang et al., 2018; Hara et al., 2017)) on the labeled cohort to predict ten binary cardiac phenotypes directly from non-cardiac chest CT. To leverage the learned alignment, we froze the lower convolutional blocks and jointly optimized the remaining layers and task-specific heads with a multi-task, masked binary cross-entropy loss, reporting AUROC for all targets.

Table 1: Performance (AUROC; mean $\pm$ standard deviation) by label on the test set under different training strategies (**bolded** values indicate best result; underlined values indicate second-best).

| Label | No pre-training | MMCL | SimCLR | NW+MMCL |
|---|---|---|---|---|
| LVEF $\leq$ 45% | $\underline{0.73} \pm 0.036$ | $\underline{0.73} \pm 0.027$ | $0.58 \pm 0.023$ | $\mathbf{0.76} \pm 0.005$ |
| LVWT $\geq$ 13 flag | $0.65 \pm 0.015$ | $\underline{0.67} \pm 0.014$ | $0.60 \pm 0.012$ | $\mathbf{0.71} \pm 0.009$ |
| Aortic Stenosis | $0.70 \pm 0.032$ | $\underline{0.74} \pm 0.022$ | $0.55 \pm 0.020$ | $\mathbf{0.85} \pm 0.042$ |
| Aortic Regurgitation | $0.63 \pm 0.020$ | $\underline{0.64} \pm 0.018$ | $0.58 \pm 0.023$ | $\mathbf{0.72} \pm 0.015$ |
| Mitral Regurgitation | $\mathbf{0.73} \pm 0.032$ | $0.72 \pm 0.014$ | $0.57 \pm 0.025$ | $\underline{0.72} \pm 0.017$ |
| Tricuspid Regurgitation | $\mathbf{0.72} \pm 0.015$ | $\underline{0.70} \pm 0.017$ | $0.62 \pm 0.020$ | $\mathbf{0.72} \pm 0.021$ |
| Pulmonary Regurgitation | $0.69 \pm 0.032$ | $\underline{0.76} \pm 0.028$ | $0.57 \pm 0.008$ | $\mathbf{0.77} \pm 0.027$ |
| PASP $\geq$ 45 flag | $\mathbf{0.68} \pm 0.015$ | $\underline{0.66} \pm 0.005$ | $0.61 \pm 0.019$ | $\mathbf{0.68} \pm 0.017$ |
| TR$_{\max}$ $\geq$ 32 flag | $\mathbf{0.67} \pm 0.020$ | $0.63 \pm 0.022$ | $0.62 \pm 0.027$ | $\underline{0.66} \pm 0.018$ |
| SHD flag | $0.70 \pm 0.032$ | $\underline{0.73} \pm 0.009$ | $0.62 \pm 0.022$ | $\mathbf{0.76} \pm 0.014$ |
| **Average** | $0.69 \pm 0.022$ | $\underline{0.70} \pm 0.012$ | $0.59 \pm 0.012$ | $\mathbf{0.73} \pm 0.007$ |

## 4. Experiments and Results

### 4.1. Dataset Overview

We assembled a large-scale multimodal cardiac imaging dataset from Columbia University Irving Medical Center and Weill Cornell Medicine comprising non-gated chest CT, ECHO, ECG, and demographic data. CT studies were temporally matched to corresponding ECHO and ECG exams using unique patient identifiers, yielding a pre-training cohort of 20,574 patients with non-contrast chest CT, ECHO, and ECG acquired within six months of the CT scan. This cohort was split at the patient level into 16,459/4,115 patients (80/20) for training and validation, and used to pre-train the cross-modal alignment and phenotype prediction framework in Section 3. For supervised downstream training, we curated a separate labeled cohort of 7,553 patients with 16,357 chest CT studies annotated for cardiac structural and functional abnormalities derived from ECHO. We defined ten binary cardiac targets, including reduced LVEF ($\leq 45\%$), increased LV wall thickness ($\geq 13$ mm), valve stenosis/regurgitation (aortic, mitral, tricuspid, pulmonary), elevated PASP ($\geq 45$ mmHg), increased $TR_{max}$ ($\geq 32$ m/s), and structural heart disease (SHD) presence (Poterucha et al., 2025). The fine-tuning cohort was split 80/20 into training and validation subsets, with an independent test set of 2,266 patients (4,861 CT studies) held out for final evaluation.

### 4.2. Implementation Details

Training and validation were conducted using PyTorch on NVIDIA A100 GPUs with mixed-precision optimization. All CT volumes were resampled to 2 mm isotropic resolution and center-cropped to $164^3$ voxels. Both the image encoder and tabular encoder were randomly initialized (Xavier for all linear layers) and trained end-to-end during pre-training. Tabular features were standardized per column. Missing entries in $\mathbf{x}_i^{tab}$, were then imputed with a k-nearest neighbors (k = 5) imputer fitted on the training split and subsequently applied to the validation and test splits. For contrastive pre-training, batch size was set to 12 with temperature $\tau_{clip}$=0.07. The Nadaraya–Watson head was initialized with $K$=20 exemplars per phenotype and refreshed every $M$=5 epochs. All reported metrics were computed on held-out patient-level splits to ensure independence across train, validation, and test cohorts.

### 4.3. Results

Table 1 summarizes performance across ten cardiac prediction tasks on the held-out chest CT test set. Our proposed framework (NW+MMCL) consistently outperformed both no-pre-training and standard multimodal contrastive learning approaches (Hager et al., 2023; Chen et al., 2020) across most of the prediction tasks (Table 1, statistical significance analysis in Appendix F). Here, one of the baselines, MMCL denotes an ablation of X-Cardia without the NW Head. Our method improved AUROC by $0.03-0.08$ over the no-pre-training baseline, except for mitral regurgitation and $TR_{max}$ ($\geq 32$ m/s). Improvements were also observed relative to standard MMCL, particularly in valvular disease tasks where physiological phenotypes yield clearer cross-modal correspondence. In the few-shot evaluation (Table 2), NW+MMCL exhibited the largest relative gains, achieving an average improvement of 11.8% AUROC over MMCL and 23.5% AUROC over SimCLR. Few-shot

gains were most pronounced for Aortic Stenosis and SHD Flag, demonstrating the effectiveness of phenotype-guided pre-training for detecting structural abnormalities. Additional experiments varying the fraction of labeled CT studies used for fine-tuning (Appendix C, Table C.1) show that NW+MMCL consistently outperforms the baseline across all data regimes, highlighting its strong sample efficiency. We also evaluate X-Cardia under zero, one, and two-shot supervision to assess representation quality and data efficiency (Appendix J). Lower absolute performance in the zero-shot setting is expected, as many cardiac phenotypes manifest subtly and are not directly observable on non-gated chest CT. With limited supervision, performance improves substantially, and X-Cardia consistently outperforms other methods across tasks. These results indicate that phenotype-guided multimodal alignment yields CT representations that generalize and adapt effectively with minimal downstream supervision.

Table 2: Test performance (AUROC; mean $\pm$ standard deviation) by label across different training strategies in 5-shot learning. (**bolded** values indicate best result; underlined values are second-best).

| Label | No pre-training | MMCL | SimCLR | NW+MMCL |
|---|---|---|---|---|
| LVEF $\leq$ 45% | $0.54 \pm 0.033$ | $\underline{0.64} \pm 0.066$ | $0.51 \pm 0.027$ | $\mathbf{0.68} \pm 0.049$ |
| LVWT $\geq$ 13 flag | $0.53 \pm 0.019$ | $\underline{0.59} \pm 0.033$ | $0.51 \pm 0.045$ | $\mathbf{0.63} \pm 0.039$ |
| Aortic Stenosis | $0.54 \pm 0.015$ | $\underline{0.64} \pm 0.026$ | $0.54 \pm 0.045$ | $\mathbf{0.86} \pm 0.018$ |
| Aortic Regurgitation | $0.53 \pm 0.025$ | $\underline{0.58} \pm 0.037$ | $0.52 \pm 0.037$ | $\mathbf{0.64} \pm 0.049$ |
| Mitral Regurgitation | $0.57 \pm 0.029$ | $\underline{0.60} \pm 0.051$ | $0.49 \pm 0.026$ | $\mathbf{0.67} \pm 0.005$ |
| Tricuspid Regurgitation | $0.53 \pm 0.025$ | $\underline{0.58} \pm 0.017$ | $0.48 \pm 0.033$ | $\mathbf{0.65} \pm 0.022$ |
| Pulmonary Regurgitation | $0.61 \pm 0.053$ | $\underline{0.67} \pm 0.029$ | $0.52 \pm 0.037$ | $\mathbf{0.69} \pm 0.008$ |
| PASP $\geq$ 45 flag | $\underline{0.55} \pm 0.026$ | $\underline{0.55} \pm 0.025$ | $0.50 \pm 0.016$ | $\mathbf{0.58} \pm 0.026$ |
| $TR_{max} \geq$ 32 flag | $\underline{0.61} \pm 0.017$ | $0.52 \pm 0.041$ | $0.53 \pm 0.015$ | $\mathbf{0.62} \pm 0.017$ |
| SHD flag | $0.59 \pm 0.035$ | $\underline{0.63} \pm 0.046$ | $0.52 \pm 0.041$ | $\mathbf{0.75} \pm 0.019$ |
| **Average** | $0.56 \pm 0.013$ | $\underline{0.60} \pm 0.013$ | $0.52 \pm 0.013$ | $\mathbf{0.68} \pm 0.002$ |

Table 3: Phenotype prediction performance with NW head versus standard linear head.

| Evaluation Metric | NW Head | Linear Head |
|---|---|---|
| AUROC | 0.66 | 0.63 |
| F1 Score | 0.60 | 0.55 |
| Cosine Similarity | 0.72 | 0.76 |

### 4.4. Ablation Study

We conducted ablations to evaluate the contributions of, (i) training with NW head vs. Linear head, (ii) the quality of cross-modal embedding alignment, and (iii) the interpretability of the support-bank representations. (iv) comparison of pre-training on cardiology vs radiology.

**NW Head.** Empirically, Table 3 shows that a learnable linear head on the fused embeddings underperforms the non-parametric, parameter free NW head on phenotype prediction,

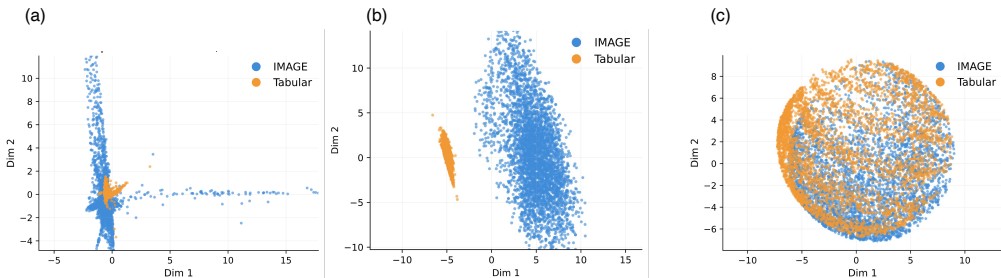

Figure 2: PCA visualization of image (blue) and tabular (orange) modality embeddings under different pre-training strategies. (a), (b), and (c) are SimCLR, MMCL, and NW + MMCL, respectively. SimCLR and MMCL show weak alignment whereas, NW + MMCL enhances alignment by integrating phenotype-level supervision, promoting structured semantic consistency across modalities and reducing modality gaps.

even though it achieves similar or slightly higher global cosine similarity. This suggests that the NW objective promotes more discriminative, phenotype-aligned decision boundaries: because the NW head has no learnable parameters, the encoders themselves must adapt to align CT and tabular embeddings, whereas a parametric head can partially bypass one modality by relying more heavily on the cleaner signal. Qualitative Grad-CAM comparisons between the NW head and a linear head further support this effect, with the NW head producing more focused cardiac attention maps (Appendix D, Figure D.1).

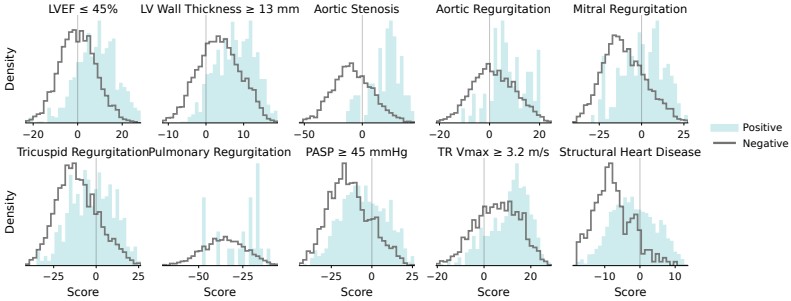

Figure 3: Task-specific score distributions for multimodal CT representations using proposed NW+MMCL based pre-training. For each task, we plot the distribution of signed classifier scores for positive (teal, filled) and negative (gray, outlined) cases, where the score denotes the signed distance of CT embeddings to the task-specific linear classifier head. Across tasks, positive cases are consistently right-shifted relative to negatives, indicating task-aligned separation, while overlap reflects clinical heterogeneity and continuous disease severity.

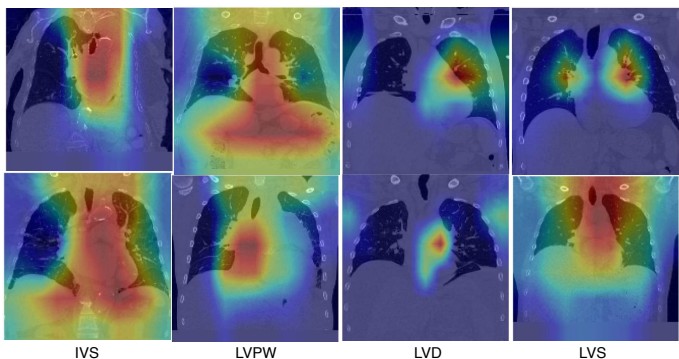

Figure 4: Qualitative Grad-CAM maps on cardiac CT slices from the final support set after cross-modal pre-training. Each column corresponds to one phenotype: IVS (interventricular septal thickness), LVPW (left ventricular posterior wall thickness), LVD (left ventricular internal diameter in diastole), and LVS (left ventricular internal diameter in systole), and each image shows a different case from the final support set. The pre-trained encoder attends to anatomically relevant cardiac regions, indicating transferable structural priors before downstream fine-tuning.

**Quality of embedding alignment.** PCA visualization (Figure 2) demonstrates that only the NW + MMCL pre-training strategy yields substantial embedding alignment between chest CT and tabular features (ECHO and ECG), forming a coherent cross-modal latent space. In contrast, SimCLR and MMCL exhibit weak or partial alignment, highlighting the necessity of phenotype-level supervision to bridge modality-specific representation gaps. These results support our hypothesis that integrating physiological phenotypes enables more effective cross-modal embedding fusion. To further assess phenotype-level discriminability in the fused multimodal space, we analyze LDA (Linear Discriminant Analysis) score distributions for key cardiac phenotypes (Appendix G). While partial overlap is observed, the consistent separation trends suggest that the fused embedding encodes clinically relevant information despite inherent phenotype variability. To evaluate whether the shared CT representation encodes task-relevant information, we examine task-specific classifier score distributions (Figure 3, Appendix G.1, and Appendix G.2). Across tasks, positive cases consistently exhibit higher scores than negatives, indicating that the NW+MMCL based pre-trained CT model captures better class-aligned evidence along task-specific directions. Although the distributions overlap, this rightward shift of positives demonstrates meaningful task-specific separation in the learned CT representation.

**Interpretability of the support-bank representations.** Support examples formed coherent manifolds with smooth phenotype gradients, and Grad-CAM maps (Figure 4) show that the pre-trained CT encoder focuses on cardiac regions despite non-cardiac, non-gated scans, indicating successful multimodal transfer of cardiac information, though chamber diameters remain harder to predict than wall thickness. Although samples within the same

phenotype class share a common label, perfect intra-class spatial consistency is not expected in this setting. Phenotypes correspond to continuous anatomical measurements whose spatial manifestations vary across patients due to differences in cardiac morphology, disease severity, and imaging plane, particularly in non-gated, non-contrast chest CT. Additional variability arises because the Nadaraya–Watson support bank aggregates multiple representative exemplars per phenotype rather than enforcing a single canonical template, encouraging the model to attend to a range of anatomically plausible regions. Consequently, Grad-CAM visualizations may exhibit heterogeneous yet phenotype-consistent attention patterns within a class, reflecting clinically meaningful variability rather than instability in the learned representation.

**Pre-training on cardiology vs radiology** We compared our cardiac modality–based pre-training approach against a CT model pretrained on radiology reports using a CLIP-style framework (Hamamci et al., 2024). During fine-tuning, the CT encoder was kept frozen while only task-specific heads were optimized. Both models were evaluated on the same ECHO-derived labels using CT volumes as input. As shown in Figure 5, our proposed method (NW+MMCL) consistently outperforms CT-CLIP across all ten downstream tasks, achieving higher AUROC for cardiac labels. The largest performance gains are observed for valvular disease classification, where cardiac-specific multimodal pre-training leveraging ECHO and ECG signals provides substantial improvements over radiology-based pre-training. These results suggest that cardiac-focused multimodal alignment yields more informative and task-relevant representations for ECHO-derived clinical variables than generic CT–report pre-training.

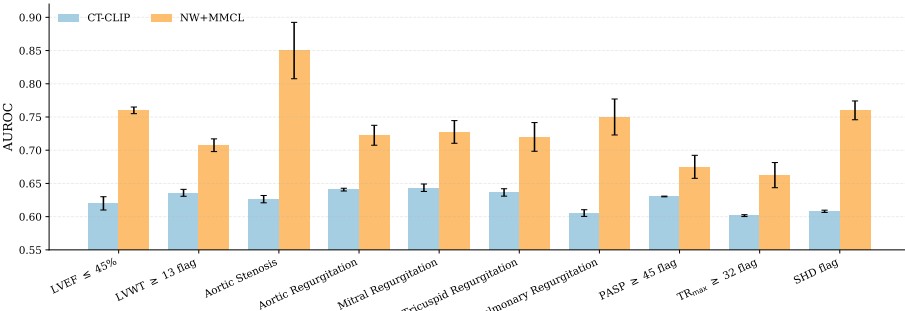

Figure 5: AUROC performance of CT-CLIP and NW+MMCL across ten cardiovascular prediction tasks on the test set. NW+MMCL consistently outperforms CT-CLIP across all tasks, with particularly large gains observed for valvular disease classification tasks. Error bars denote variability across repeated runs (standard deviation).

### 4.5. Discussion and Limitations

Our proposed phenotype-guided multimodal alignment framework demonstrates several significant advantages. By enforcing consistency between cardiac measurements and CT-

derived embeddings during pre-training, the model learns physiologically relevant representations that transfer effectively to non-gated chest CT. This leads to strong gains across all cardiac prediction tasks and is especially beneficial in data-scarce scenarios, where our approach consistently outperforms standard contrastive pre-training and no pre-training baselines.

Additionally, the non-parametric Nadaraya–Watson head serves as a structural safeguard against modality collapse. In contrast to parametric classifiers that can implicitly down-weight the noisier signal in favor of cleaner features, the NW head has no learnable parameters and therefore cannot internalize the supervision itself. This places the optimization burden entirely on the encoders, encouraging the CT backbone to shape its embedding geometry to align with the phenotype-support prototypes. The exemplar-based formulation also offers a more transparent link between predictions and representative cases, helping to anchor the latent space in clinically meaningful phenotypes. Notably, despite the CT scans not being acquired for cardiac indications, the pre-trained model learns to focus on cardiac regions, suggesting effective transfer of structural information from ECHO and ECG into CT-based representations.

Despite these strengths, several limitations should be considered. The multimodal alignment depends on exams occurring within six months of the CT scan, during which a patient's cardiac condition may change, potentially introducing misalignment. Phenotype labels were derived using threshold-based binarization, which may simplify complex physiological conditions. Furthermore, the current support bank does not dynamically adapt to rare or evolving phenotypes. Finally, our evaluation focused on binary classification tasks; future extensions could explore continuous severity prediction, temporal modeling for deeper clinical insight. Additionally, paired CT–ECG–ECHO data are required only during pre-training as a one-time supervision cost and are not needed at inference, where the model operates solely on CT, enabling scalable deployment for opportunistic cardiac screening. Because retrospectively collected multimodal cohorts may represent a clinically enriched population, their distribution may differ from that of patients undergoing routine CT alone; future work will investigate bias-aware pre-training and domain adaptation strategies to mitigate this potential distribution shift. Future extensions could also explore more expressive image encoder architectures to further improve representation capacity and downstream performance.

## 5. Conclusion

This work presents a phenotypically supervised multimodal alignment framework that unifies cardiac and non-cardiac imaging by leveraging tabular cardiac features as alignment signals. By integrating CLIP-style cross-modal contrastive learning and a Nadaraya–Watson non-parametric head, the method produces semantically grounded, interpretable embeddings that transfer effectively to chest CT for cardiac abnormality prediction. Extensive experiments demonstrate consistent improvements across ten cardiac tasks, strong data efficiency, and substantial few-shot gains. Together, these results highlight phenotype-level alignment as a promising direction for multimodal representation learning in medical imaging, particularly when labeled data are scarce or modalities differ in diagnostic intent.

## Acknowledgments

This work was fully supported by funding from NewYork-Presbyterian for the NYP-Cornell Cardiovascular AI Collaboration. We gratefully acknowledge the contributions of the data team and the clinicians involved in this work.

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

## Appendix A. Test set overview

Table A.1: Label distribution and missingness rate in the test cohort across ten cardiac abnormalities. Counts represent positive (pos), negative (neg), and rate of missing labels per class.

| Label | Pos | Neg | Missing (%) |
|---|---|---|---|
| LVEF $\leq 45\%$ | 553 | 4304 | 0.1 |
| LVWT $\geq 13$ flag | 624 | 3691 | 11.2 |
| Aortic stenosis | 126 | 3470 | 26.0 |
| Aortic regurgitation | 75 | 4575 | 4.3 |
| Mitral regurgitation | 196 | 4428 | 4.9 |
| Tricuspid regurgitation | 363 | 4267 | 4.8 |
| Pulmonary regurgitation | 15 | 3334 | 31.1 |
| PASP $\geq 45$ flag | 886 | 1920 | 42.3 |
| $TR_{max} \geq 32$ flag | 422 | 1733 | 55.7 |
| SHD flag | 1942 | 170 | 56.6 |

## Appendix B. Effect of NW Head Loss Weighting ($\lambda_{\text{nw}}$)

Varying the NW loss weight (Table B.1) revealed that excessively small values underutilize phenotype structure, while very large values can overconstrain the representation. The optimal range lies between 0.4 and 0.6, balancing alignment and phenotype smoothing.

Table B.1: Performance (AUROC) on the test set by label using different $\lambda_{nw}$ values in the loss.

| Label | $\lambda_{nw} = 0.2$ | $\lambda_{nw} = 0.4$ | $\lambda_{nw} = 0.6$ | $\lambda_{nw} = 0.8$ | $\lambda_{nw} = 1.0$ |
|---|---|---|---|---|---|
| LVEF $\leq 45\%$ | 0.74 | 0.74 | 0.73 | 0.74 | 0.74 |
| LVWT $\geq 13$ flag | 0.72 | 0.69 | 0.70 | 0.69 | 0.71 |
| Aortic Stenosis | 0.89 | 0.90 | 0.87 | 0.88 | 0.88 |
| Aortic Regurgitation | 0.72 | 0.67 | 0.65 | 0.68 | 0.73 |
| Mitral Regurgitation | 0.73 | 0.73 | 0.72 | 0.72 | 0.75 |
| Tricuspid Regurgitation | 0.66 | 0.69 | 0.66 | 0.65 | 0.70 |
| Pulmonary Regurgitation | 0.81 | 0.73 | 0.77 | 0.72 | 0.66 |
| PASP $\geq 45$ flag | 0.67 | 0.68 | 0.67 | 0.68 | 0.69 |
| $TR_{max} \geq 32$ flag | 0.64 | 0.65 | 0.65 | 0.67 | 0.65 |
| SHD flag | 0.77 | 0.76 | 0.76 | 0.75 | 0.75 |

## Appendix C. Effect of Training Size

Table C.1 reports performance as the labeled training set is reduced to $2\%, 4\%, 6\%$, and $10\%$ of available CTs. NW+MMCL outperformed the baseline at every fraction and for every label. In extremely low-supervision settings (e.g., $2\%$), the method improved AUROC by up to $57\%$ for Aortic Stenosis and $48\%$ for Mitral Regurgitation than the no-pre-training baseline. These results indicate that phenotype-guided multimodal alignment yields strong sample efficiency.

Table C.1: Performance (AUROC) by disease label across models at different fractions of the training set. Best per percent per row is bolded.

| | 2% | | 4% | | 6% | | 10% | |
|---|---|---|---|---|---|---|---|---|
| Label | Base | NW+MMCL | Base | NW+MMCL | Base | NW+MMCL | Base | NW+MMCL |
| Aortic Regurgitation | 0.56 | **0.73** | 0.51 | **0.71** | 0.56 | **0.72** | 0.62 | **0.66** |
| Aortic Stenosis | 0.49 | **0.77** | 0.53 | **0.79** | 0.43 | **0.78** | 0.62 | **0.78** |
| LVEF $\leq 45\%$ | 0.58 | **0.69** | 0.61 | **0.73** | 0.54 | **0.73** | 0.50 | **0.74** |
| LVWT $\geq 13$ flag | 0.56 | **0.64** | 0.63 | **0.69** | 0.52 | **0.69** | 0.54 | **0.68** |
| Mitral Regurgitation | 0.49 | **0.73** | 0.56 | **0.74** | 0.56 | **0.73** | 0.56 | **0.71** |
| PASP $\geq 45$ flag | 0.47 | **0.65** | 0.50 | **0.67** | 0.54 | **0.66** | 0.63 | **0.67** |
| Pulmonary Regurgitation | 0.72 | **0.73** | 0.58 | **0.69** | 0.55 | **0.61** | 0.65 | **0.69** |
| SHD flag | 0.46 | **0.72** | 0.61 | **0.71** | 0.52 | **0.73** | 0.63 | **0.72** |
| TR$_{max} \geq 32$ flag | 0.46 | **0.64** | 0.51 | **0.63** | 0.52 | **0.63** | 0.62 | **0.63** |
| Tricuspid Regurgitation | 0.57 | **0.71** | 0.50 | **0.70** | 0.57 | **0.69** | 0.63 | **0.69** |

## Appendix D. Non-Parametric vs. Learnable Heads for Phenotype Prediction

In this appendix (Figure D.1), we qualitatively examine how the choice of prediction head influences the spatial focus of the pre-trained CT encoder. Grad-CAM visualizations for four key phenotypes show that the non-parametric NW head, when combined with multi-modal contrastive pre-training, consistently drives the encoder to attend to anatomically relevant myocardial and chamber regions, in line with the underlying ECHO-derived measurements. In contrast, replacing the NW head with a linear classifier yields more scattered and occasionally extra-cardiac attention patterns. These examples support our hypothesis that the non-parametric NW head better enforces phenotype-level alignment between CT and tabular representations, leading to more interpretable and physiologically grounded image features.

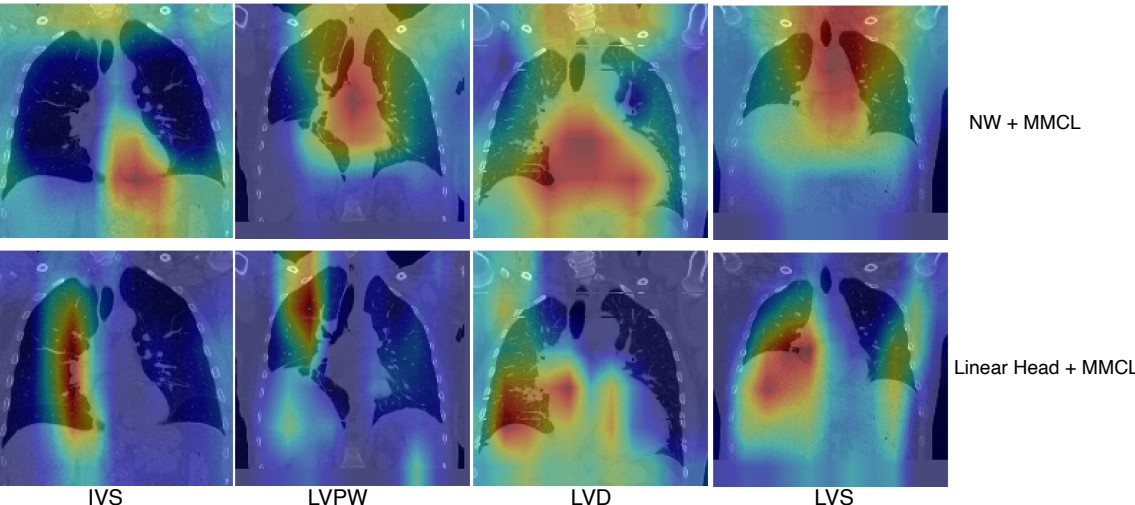

Figure D.1: Comparison of Grad-CAM attention maps on cardiac CT volumes for the non-parametric Nadaraya–Watson (NW) head versus a learnable linear head after cross-modal pre-training. Each column corresponds to one of four phenotypes—IVS (interventricular septal thickness), LVPW (left ventricular posterior wall thickness), LVD (left ventricular internal diameter in diastole), and LVS (left ventricular internal diameter in systole). The NW + MMCL model (top row) produces sharper, more localized attention over ventricular walls and chambers, whereas the Linear Head + MMCL model (bottom row) exhibits more diffuse and off-target responses, indicating weaker phenotype alignment in the learned image features.

## Appendix E. Embedding Combination Strategies

We ablate three strategies for combining image and tabular embeddings for phenotype prediction, while keeping the encoders, training schedule, and Nadaraya–Watson (NW) head fixed across settings.

**(i) Sum fusion.** **Given the image embedding $e^{\mathbf{img}} \in \mathbb{R}^d$ and tabular embedding $e^{\mathbf{tab}} \in \mathbb{R}^d$, we compute the fused representation via element-wise summation followed by $\ell_2$ normalization:**

$$z = \frac{e^{\mathbf{img}} + e^{\mathbf{tab}}}{\|e^{\mathbf{img}} + e^{\mathbf{tab}}\|_2} \tag{8}$$

**This corresponds to the fusion operation defined in Eq. (3) of the main text.**

**(ii) Concatenation.** We concatenate modality-specific embeddings and project them back to the shared embedding dimension:

$$z = \frac{[e^{\mathrm{img}}; e^{\mathrm{tab}}]}{\|[e^{\mathrm{img}}; e^{\mathrm{tab}}]\|_2} \tag{9}$$

where $[\cdot; \cdot]$ denotes concatenation. This ensures dimensional compatibility with other fusion strategies.

**(iii) Gated fusion.** We employ a learned scalar gating mechanism to adaptively weight the two modalities. Specifically, we compute a gating coefficient

$$a = \sigma\Big(g\Big([e^{\mathrm{img}}; e^{\mathrm{tab}}]\Big)\Big), \tag{10}$$

where $g(\cdot)$ is a small multilayer perceptron and $\sigma(\cdot)$ is the sigmoid function. The fused embedding is then given by

$$z = \frac{a\, e^{\mathrm{img}} + (1 - a)\, e^{\mathrm{tab}}}{\|a\, e^{\mathrm{img}} + (1 - a)\, e^{\mathrm{tab}}\|_2}. \tag{11}$$

This strategy can be interpreted as adaptive scalar-gated late fusion, related to gated multimodal fusion approaches (Arevalo et al., 2017). Table E.1 reports performance for each fusion strategy using AUROC, F1 score, and cosine similarity between predicted and ground-truth phenotypes. All results are computed on the same evaluation split using an identical support-bank construction for the NW head.

We adopt sum fusion to ensure balanced and stable training of both the image and tabular encoders, as it enforces equal contribution from each modality and avoids optimization bias toward a single encoder. Empirically, sum fusion achieves performance comparable to more complex fusion strategies across AUROC, F1, and cosine similarity (Table E.1), motivating its use as a simple and robust default.

Table E.1: Comparison of embedding combination strategies. Best performance per row is bolded.

| Evaluation Metric | Sum Fusion | Concatenation | Gated Fusion |
|---|---|---|---|
| AUROC | **0.66** | 0.65 | 0.62 |
| F1 Score | **0.60** | 0.58 | 0.58 |
| Cosine Similarity | 0.72 | 0.72 | **0.74** |

| Task | NW+MMCL vs. MMCL | | NW+MMCL vs. SimCLR | | NW+MMCL vs. No pre-training | |
|---|---|---|---|---|---|---|
| | p-value | Sig. | p-value | Sig. | p-value | Sig. |
| LVEF | 0.080 | No | $3.9 \times 10^{-4}$ | Yes | 0.314 | No |
| LVWT | 0.004 | Yes | $2.2 \times 10^{-4}$ | Yes | 0.044 | Yes |
| Aortic stenosis | 0.026 | Yes | 0.0077 | Yes | 0.050 | Yes |
| Aortic regurgitation | 0.024 | Yes | 0.0011 | Yes | 0.009 | Yes |
| Mitral regurgitation | 0.215 | No | $4.9 \times 10^{-4}$ | Yes | 0.286 | No |
| Tricuspid regurgitation | 0.102 | No | $5.8 \times 10^{-4}$ | Yes | 0.667 | No |
| Pulmonary regurgitation | 0.239 | No | 0.0010 | Yes | 0.081 | No |
| PASP | 0.141 | No | 0.0022 | Yes | 0.226 | No |
| $TR_{max}$ | 0.030 | Yes | 0.030 | Yes | NA | NA |
| SHD | 0.062 | No | 0.0021 | Yes | 0.184 | No |
| **Macro-average** | **0.020** | **Yes** | $3.1 \times 10^{-5}$ | **Yes** | **0.037** | **Yes** |

Table F.1: p-values from paired statistical tests. We report p-values from paired two-sided t-tests comparing NW+MMCL against each strategies. "Sig." denotes statistical significance at $p < 0.05$. The macro-average row corresponds to a paired test on run-wise macro-averaged AUROC (averaged across all tasks within each run). "NA" indicates a degenerate case with zero variance in paired differences.

## Appendix F. Statistical Significance Analysis

We assessed statistical significance using paired, two-sided t-tests across repeated runs with identical experimental settings. For each tasks, AUROC values from the same run were treated as paired observations. A significance level of $\alpha = 0.05$ was used. Table F.1 reports the p-values and significance outcomes for comparisons with approaches.

NW+MMCL shows statistically significant improvements in macro-averaged AUROC over all other strategies, with particularly strong gains relative to SimCLR and training from scratch. Several individual tasks do not reach significance when compared to MMCL, reflecting the limited number of paired runs rather.

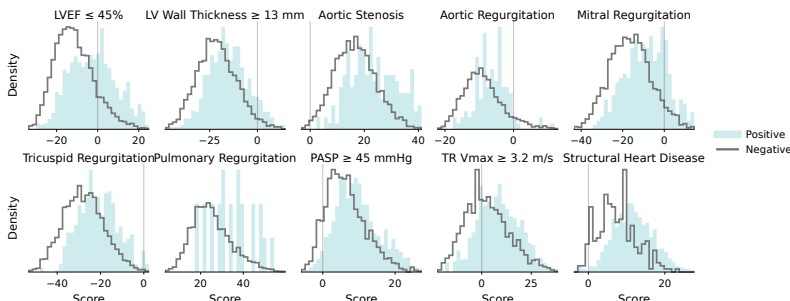

Figure G.1: Task-specific score distributions for multimodal CT representations using MMCL pre-training. For each task, we plot the distribution of signed classifier scores for positive (teal, filled) and negative (gray, outlined) cases, where the score denotes the signed distance of CT embeddings to the task-specific linear classifier head.

## Appendix G. Discriminative Analysis of Features

Relative to MMCL (Figure G.1) and SimCLR (Figure G.2), the NW+MMCL model (Figure 3) produces more consistent right-shifted score distributions for positive cases across tasks, indicating improved task-aligned separation. MMCL shows moderate but less stable separation, while SimCLR exhibits substantial overlap, highlighting the benefit of phenotype-guided multimodal pre-training. On the other hand, to quantify phenotype-level separability in the fused multimodal representation using NW+MMCL pre-training, we projected the latent embeddings onto a one-dimensional LDA axis for each concept and visualized the resulting score distributions using kernel density estimates (Figure G.3). Unlike unsupervised dimensionality reduction methods, this approach directly measures how well the fused embedding linearly separates negative (in-range) and positive (out-of-range) samples. Across all four concepts (IVS, LVPW, LVD, and LVS), the fused representation exhibits systematic shifts in the LDA score distributions, with varying degrees of overlap between classes. Phenotypes with more clearly separated distributions demonstrate stronger alignment, whereas increased overlap suggests reduced discriminability. Overall, these results indicate that the fused embedding captures relevant structure and supports interpretable assessment of multimodal feature alignment at the phenotype level.

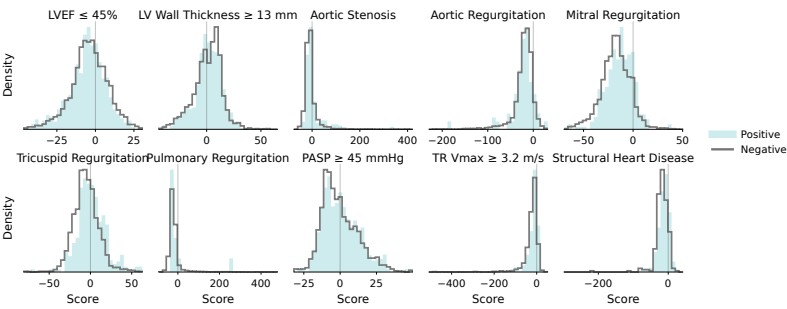

Figure G.2: Task-specific score distributions for multimodal CT representations using Sim-CLR pre-training. For each task, we plot the distribution of signed classifier scores for positive (teal, filled) and negative (gray, outlined) cases, where the score denotes the signed distance of CT embeddings to the task-specific linear classifier head.

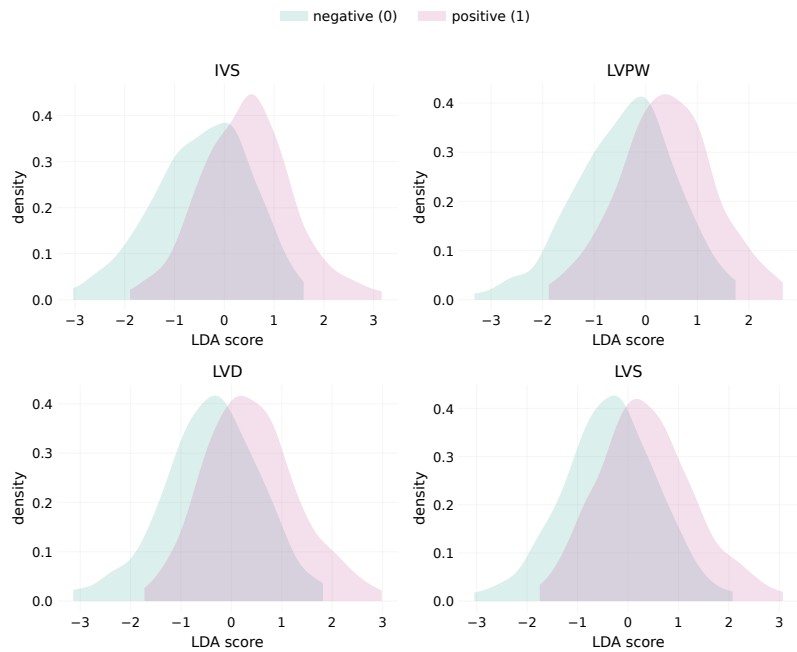

Figure G.3: Phenotype separability in the fused latent space. Kernel density estimates of LDA projection scores for four cardiac concepts (IVS, LVPW, LVD, LVS) derived from the fused embedding. Each panel shows the one-dimensional LDA axis that maximizes class separation between negative and positive samples. Clear shifts between distributions indicate stronger phenotype discriminability in the fused representation.

## Appendix H. Tabular Feature Specification and Embedding Construction

This appendix provides details on the structured ECG and ECHO features used to construct the tabular embeddings in X-Cardia, including feature composition, embedding dimensionality, and feature selection considerations.

### H.1. ECG and ECHO Feature Set

The tabular input consists of routinely reported clinical measurements derived from ECHO, ECG, and basic demographics. These features reflect standard cardiac structure, function, and conduction parameters commonly available in clinical workflows. Specifically, the structured feature set includes:

- **Demographics and clinical context:** patient age, sex, race, ventricular assist device (VAD) flag.

- **ECG-derived measurements:** ventricular rate, atrial rate, PR interval, QRS duration, QT interval, corrected QT (QTc), ventricular pacing flag.

- **Cardiac function and pressure estimates:** left ventricular ejection fraction (LVEF), pulmonary artery systolic pressure (PASP), PASP excluding right atrial pressure, right atrial pressure (RAP), pericardial effusion indicator, tricuspid regurgitation maximum velocity.

- **Diastolic function parameters:** E-wave velocity, A-wave velocity, E/A ratio, mitral valve tissue Doppler indices (E$'$ and A$'$ velocities: lateral, medial, and averaged).

- **Left ventricular outflow tract (LVOT) measurements:** LVOT area, diameter, peak and mean velocities, velocity–time integral (VTI), peak and mean pressure gradients.

- **Valvular disease and prosthesis indicators:** global and valve-specific prosthetic flags (aortic, mitral, tricuspid, pulmonary), mitral regurgitation VTI and peak velocity.

- **Aortic valve hemodynamics:** aortic valve peak velocity, peak gradient, mean gradient, and VTI.

### H.2. Tabular Embedding Dimensionality

All structured features are standardized and encoded using an FT-Transformer based tabular encoder. The resulting tabular embedding is projected into a shared latent space of dimension, $d = 256$, where both CT and tabular embeddings are aligned and fused via element-wise summation.

### H.3. Feature Selection Considerations

We do not perform explicit feature selection or subset optimization. This design choice is intentional, as the tabular encoder is used to provide physiologically meaningful intermediate supervision rather than to optimize standalone tabular prediction. Using a broad set of

| Parameter | Value |
|---|---|
| Epochs | 30 |
| Batch size | 8 |
| Optimizer | AdamW |
| Learning rate | $1 \times 10^{-4}$ |
| Weight decay | $1 \times 10^{-5}$ |
| LR scheduler | CosineAnnealingWarmRestarts |
| Scheduler params | $T_0{=}10$, $T_{\mathrm{mult}}{=}2$, $\eta_{\min}{=}1 \times 10^{-6}$ |
| CLIP temperature ($\tau$) | 0.07 |
| NW softmax temperature ($\tau_{\mathrm{NW}}$) | 1.0 |
| NW loss weight ($\lambda_{\mathrm{NW}}$) | 0.5 |
| Support size cap per concept ($K$) | 5 |
| Support refresh interval | every 5 epochs |
| K-means ($k$) for support selection | up to 5 |

Table I.1: Pre-training hyperparameters used in all experiments.

| Parameter | Value |
|---|---|
| Epochs (max) | 100 |
| Batch size | 12 |
| Optimizer | AdamW |
| Learning rate | $5 \times 10^{-4}$ |
| Weight decay | $1 \times 10^{-3}$ |
| LR scheduler | OneCycleLR (cosine anneal) |
| Early stopping patience | 6 |

Table I.2: Fine-tuning hyperparameters and evaluation protocol.

routinely available cardiac measurements encourages the CT encoder to align with diverse structural, functional, and hemodynamic signals during multimodal pre-training. Exploring learned or task-adaptive feature selection remains an interesting direction for future work.

## Appendix I. Additional Experimental Details: Hyperparameters, Selection, and Compute

### I.1. Hyperparameters

We added the details of hyperparameters used in Table I.1 and Table I.2.

### I.2. Selection Criteria for Key Parameters

- **K-means support selection ($k$).** We cap the number of support exemplars per phentype with $K{=}5$ to balance diversity and retrieval cost. For phenotype $p$ and number of positive cases $N_p^+$, we set $k = \min(K, N_p^+)$; when $N_p^+ \leq K$, we use all positives.

| Setting | Value |
|---|---|
| GPU | NVIDIA A100 40GB |
| GPUs used | 2 |
| Pre-training time / epoch | $\sim 55$ min |
| Fine-tuning time / epoch | $\sim 40$ min |
| Precision | FP32 |

Table I.3: Hardware and wall-clock training time.

- **Contrastive temperature ($\tau$).** We use $\tau=0.07$ (CLIP-style default), which was stable across runs; we did not observe meaningful gains from additional tuning in our setting.

- **Support refresh interval.** Updating the support bank every 5 epochs provided a good trade-off between adaptivity and overhead.

### I.3. Compute and Complexity

We report wall-clock training time for pre-training and fine-tuning, along with GPU type and number of GPUs used, in the final version. Overall computational cost is dominated by the 3D CNN backbone, while the additional overhead from multimodal fusion and support-set retrieval is minimal due to the small, fixed-size support bank.

## Appendix J. Additional Results and Robustness Analysis

**Few-shot evaluation.** We report performance under $K$-shot settings ($K \in \{0, 1, 2\}$). In the 0 shot case, the pre-trained CT encoder is evaluated with a frozen backbone. For 1 and 2 shot settings, the model is fine-tuned using $K$ labeled samples per class with fixed data splits across methods. NW+MMCL consistently outperforms other methods across tasks and shot settings. Performance improves with increasing supervision for all methods; however, NW+MMCL shows steeper gains, indicating better sample efficiency and more transferable CT representations.

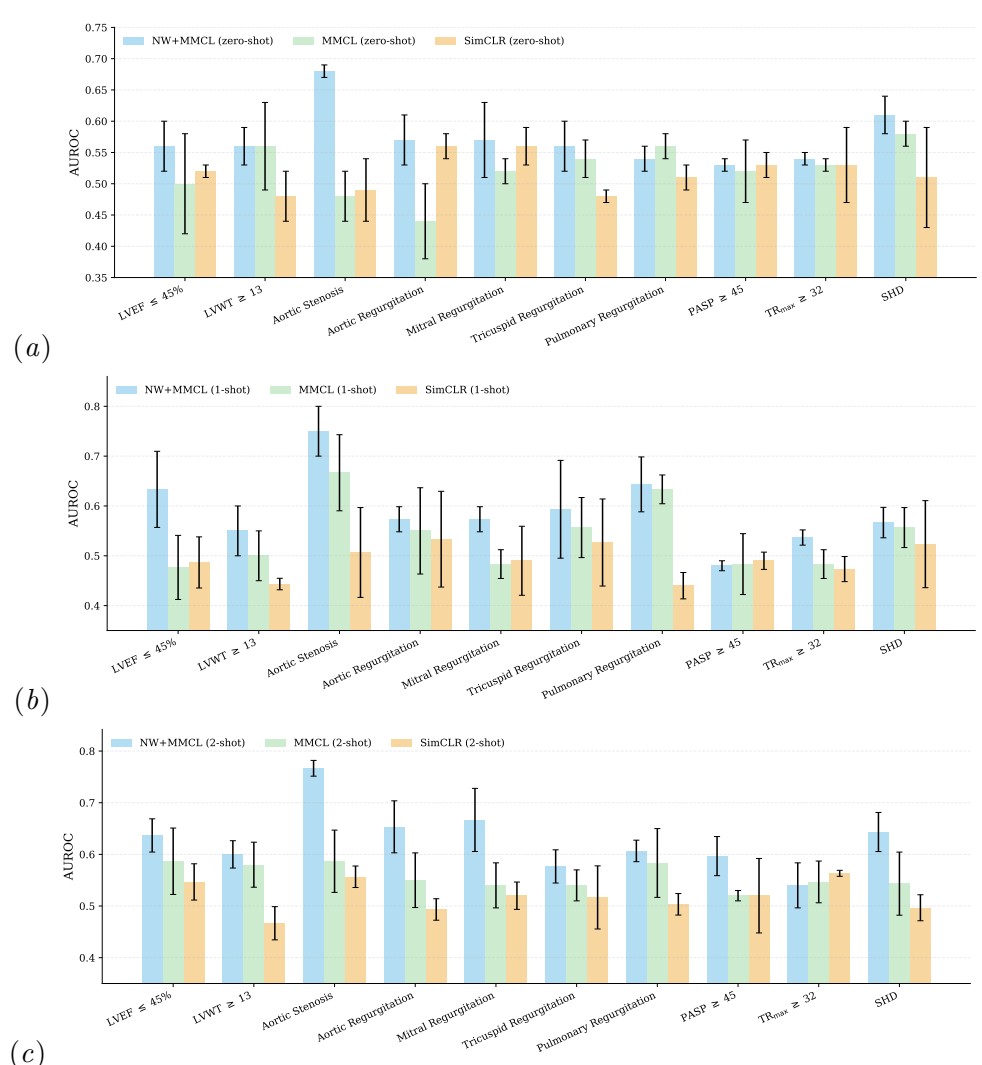

Figure J.1: Few-shot downstream performance across cardiac tasks. AUROC (mean ± std over repeated runs) for 10 cardiac phenotype prediction tasks under (a) 0 shot, (b) 1 shot, and (c) 2 shot supervision.

