# OpenReview forum: "X-Cardia: Phenotype-Guided Cross-Modal Alignment for Opportunistic Cardiac Screening on Routine Chest CT"
_MIDL.io/2026/Conference — MIDL 2026 Poster_

### Official Review · Reviewer_apw5 · 2026-01-03

**Confidence:** 4
**Preliminary Rating:** 4
**Final Rating:** 4

**Summary:**

This paper proposes a classification regime trained using contrastive objectives for abnormalities classification based on CT scans and ECHO/ECG features. The pipeline fuses multimodal features to perform cross-modal alignment, permutating both imaging features and tabular features to construct contrastive learning samples. Detection results are good with paired Grad-CAM visualizations.

**Strengths:**

1. The organization is clear, and the paper is easy to understand.

2. Using CT imaging modality to probe cardiovascular disease is an interesting idea.

3. Permutating on both modalities with respect to constructing contrastive learning samples is an interesting method advancing from existing single-modal/cross-modal contrastive learning formulations.

4. The results seem to be good.

**Weaknesses:**

1. Visualizations in Figure 2 might benefit from further discussions on whether latent embeddings are class-discriminative. Overall, I agree with the authors that Figure 2(c) shows that the cross-modal latent features overlap. At the same time, I believe adding discussions/additional visualizations of the class-discriminability of the latent features would benefit existing discussions, since the model is trained with contrastive-learning objectives, and high-quality features should be able to discriminate among classes.

2. Intra-class consistency issues in Figure 3. I believe the area where the model places more attention should be consistent across intra-class samples, but there are some discrepancies in some columns in Figure 3 (i.e., LVPW, LVS). Further discussion would benefit this part of the visualization.

3. Experimental details. I would like to ask for further clarification on how the experiment parameters are selected (e.g., k in k-means, temperature in pre-training), and more details of training time and computational complexity.

**Detailed Comments:**

Please refer to the weakness part.

**Justification Of Final Rating:**

I appreciate the authors' efforts to address my concerns. I keep my positive scores for the following reasons: 1. The problem investigated is very interesting. 2. The proposed methodology is novel with respect to constructing cross-modality learning samples. 2. The authors have released code to ensure reproducibility.

**Justification Of The Preliminary Rating:**

Overall, the paper is well-written, and the problems are clearly defined. Developing methods for cardiac screening using challenging imaging modalities is an interesting avenue, and using permutations of both modalities when constructing learning samples is an interesting approach for multi-modal alignment. This paper could benefit from further clarification of details and more discussion of existing visualizations.

**Questions To Address In The Rebuttal:**

Please refer to the weakness part.

---

> ### Author Response · Authors · 2026-01-24
>
> We thank the Reviewer apw5 for these detailed comments, and would like to respond to the questions:
>
> ### **Q1: Class-discriminability of latent embeddings**
>
> **Response:** We thank the reviewer for this constructive suggestion. We agree that **Figure 2** primarily illustrates cross-modal alignment rather than class-level separation, and that overlap in the shared latent space is expected given the continuous and heterogeneous nature of cardiac phenotypes. To address class-discriminability explicitly, we include additional comparative analyses that examine task and phenotype-specific separation in the learned representations, including task-specific classifier score distributions and LDA-based projections in **Figure 3** and **Appendix G**. These results show consistent directional separation between positive and negative cases despite overlap using the NW+MMCL pre-training method, indicating that the aligned latent space encodes class-relevant information while preserving physiologically continuous structure. We further clarify in the revised manuscript (highlighted in red, **Section 4.4**) how alignment and discriminability play complementary roles in shaping task-relevant and transferable representations.
>
> ### **Q2: Intra-class variability in Grad-CAM visualizations**
>
> **Response:** We thank the reviewer for highlighting this point. We clarify that phenotypes such as LVPW and LVS correspond to continuous anatomical measurements whose spatial manifestations vary across patients due to morphology, disease severity, and imaging plane, particularly in non-gated CT. Additionally, the non-parametric support-bank design aggregates multiple exemplars rather than enforcing a single canonical pattern. A **clarifying discussion** (Section 4.4) has been added to explain why such variability reflects clinically meaningful heterogeneity rather than model instability.
>
> ### **Q3: Experimental details and computational cost**
>
> **Response:** Thank you for raising this point. We have added a **brief discussion of computational cost and training efficiency** in the revised manuscript (Appendix I). Specifically, we now report wall-clock training time, GPU type, and number of GPUs used for both pre-training and fine-tuning. We also clarify that the overall computational complexity is dominated by the 3D CNN backbone, while the additional overhead from multimodal fusion and the Nadaraya–Watson support-set retrieval is minimal due to the small, fixed-size support bank.

---

### Official Review · Reviewer_Dgqs · 2026-01-07

**Confidence:** 5
**Preliminary Rating:** 4
**Final Rating:** 4

**Summary:**

This work pre-trains a backbone model on CT volumes, uses CLIP to align the representation with tabular data, and also adds a phenotype prediction loss on the embeddings.
......................................................................................................................................................

**Strengths:**

This paper aims to tackle multi modal integration across many different data sources to improve phenotype prediction.

.............................................................................................................................

**Weaknesses:**

The results don't have standard deviations so it is unclear if the results are significant or random chance. The "no pre-training" and "SimCLR" appear almost at a random chance AUC which is strange.
The data and code don't appear to be public, limiting the ability to reproduce the results.

**Detailed Comments:**

It would be useful to produce results on an open dataset so other work can compare against this method.

**Justification Of Final Rating:**

..........................................................................................................................................................................................................

**Justification Of The Preliminary Rating:**

The work appears correct, while the impact is likely limited due to data and code not being available the method is interesting and will be of interest to the community.
....................................

**Questions To Address In The Rebuttal:**

See above

---

> ### Author Response · Authors · 2026-01-24
>
> We thank the Reviewer Dgqs for these detailed comments, and would like to respond to the questions:
>
> ### **Q1: Lack of standard deviations and near-random baseline performance**
>
> **Response:** We apologize for the lack of clarity in the initial submission. We now report **mean ± standard deviation** for all AUROC results and include **statistical significance testing** (Appendix F), confirming that the reported gains are consistent across runs. The near-random performance of SimCLR and no-pretraining baselines reflects the intrinsic difficulty of predicting cardiac phenotypes from non-gated, non-contrast chest CT without phenotype-guided supervision.
>
> ### **Q2: Dataset and code availability**
>
> **Response:** Due to the presence of protected health information (PHI) and IRB restrictions, the dataset cannot be publicly released. To support reproducibility and facilitate future research, we are releasing the full codebase and pre-trained model weights upon publication.

---

> > ### Author Response · Authors · 2026-01-28
> >
> > ### **Q3: Open dataset performance**
> >
> > **Response:** We agree that evaluation on an open dataset would facilitate broader comparison. However, most publicly available CT datasets are based on contrast-enhanced or gated cardiac CT and do not provide paired ECG and ECHO measurements, which are required for the proposed phenotype-guided pre-training strategy. Due to the lack of publicly available datasets with this specific multimodal combination, direct comparison on an open dataset is not currently feasible. To support reproducibility and external evaluation, we release the full codebase and pre-trained model weights, enabling other researchers to apply our pre-training approach and assess its performance on their own datasets for cardiac abnormality detection.

---

> > > ### Comment · Reviewer_Dgqs · 2026-01-28
> > >
> > > I understand that it is hard to find datasets with all the features available. But you could evaluate on a single modality dataset to demonstrate that the CLIP loss improved representation learning on the image encoder right? If there is substantial "embedding alignment" then you wouldn't need the tabular data during inference and this would confirm the statements regarding that.

---

> > ### Author Response · Authors · 2026-01-28
> >
> > We thank the reviewer for this insightful suggestion. While evaluating on a single-modality public CT dataset could demonstrate general representation quality, the primary goal of this work is to enable opportunistic cardiac screening from routine, non-contrast, and non-gated chest CT through  **phenotype-guided pre-training using ECG and ECHO supervision**. To our knowledge, **no public dataset** provides non-contrast chest CT paired with ECHO and ECG derived cardiac phenotypes, which are required to meaningfully evaluate the proposed pre-training strategy and evaluate on cardiac abnormality predictions.
> >
> > To assess whether phenotype-guided multimodal alignment improves the CT image encoder itself, we include a CT-only comparison against a strong vision foundation baseline, CT-CLIP (used CLIP style pre-training), which is pre-trained on large-scale radiology reports. As shown in Figure 5 (revised manuscript), the CT encoder learned via our proposed method consistently outperforms CT-CLIP across downstream cardiac tasks, despite using only CT inputs at inference. This result provides empirical evidence that **phenotype-guided alignment with CLIP loss** using ECHO and ECG data, yields more cardiac-relevant and transferable representations than radiology-text pre-training using CLIP loss alone.
> >
> > Given the absence of suitable public datasets with paired non-contrast chest CT and cardiac phenotypes, we could not assess the embedding alignment for cardiac label predictions for external validation. We release our full codebase and pre-trained weights to facilitate external validation on institutional datasets where such multimodal data are available, and we view broader evaluation as an important direction for future work.

---

### Official Review · Reviewer_U9PA · 2026-01-09

**Confidence:** 5
**Preliminary Rating:** 3
**Final Rating:** 4

**Summary:**

The authors propose X-Cardia, a multimodal framework that integrates routine non-contrast, non-gated CT images with tabular measurements from echocardiography (ECHO) and electrocardiography (ECG) using CLIP-style contrastive pre-training on a cohort of 20,574 patients. A key methodological contribution is the use of a Nadaraya–Watson (NW) prediction head to encourage phenotype-level similarity across modalities. The proposed approach demonstrates notable performance gains, including an 8% AUROC improvement on held-out test sets and a 9.8% AUROC improvement in 5-shot classification for ECHO abnormality detection.

**Strengths:**

1. The study presents an effective and clinically meaningful integration of routine non-contrast, non-gated CT scans with tabular embeddings derived from ECHO and ECG, offering a simple yet powerful multimodal design.

2. The proposed X-Cardia framework employs a non-parametric Nadaraya–Watson prediction head with a cosine-similarity–based kernel and temperature scaling, which enhances robustness to noise and helps mitigate overfitting.

3. The manuscript includes robust and extensive ablation studies examining different prediction heads, alignment strategies, and the interpretability of the support bank, strengthening confidence in the proposed design choices.

**Weaknesses:**

1. The experimental evaluation does not include baseline models that rely solely on tabular ECG and ECHO measurements to predict the target ECHO labels. Including such baselines—ranging from simple statistical or linear models to more expressive tabular learners—would help establish meaningful upper and lower performance bounds and better contextualize the gains achieved by X-Cardia.

2. The scalability of the proposed approach may be constrained by the availability of paired datasets that include CT images alongside ECG and ECHO measurements. A more detailed discussion of how this requirement affects generalizability and deployment in broader clinical settings would strengthen the manuscript.

**Detailed Comments:**

1.	For all tables, it is hard to know the confidence interval or standard deviation of the AUROC.

**Justification Of Final Rating:**

The authors have shown extensive experiments to demonstrate improved zero-shot/few-shot performance on the tasks. Although the X-Cardia did not dominate across all tasks, it has shown significant improvement on a few tasks (e.g., aortic stenosis) which might be missed during the CT screening alone. This shows the importance of the multimodal (cross-modal) pre-training. I have raised the final rating.

**Justification Of The Preliminary Rating:**

1. The paper is well structured and well motivated, proposing a multimodal pre-training framework that integrates routine CT imaging with ECG and ECHO measurements. This design enriches the learned visual representations and leads to improved supervised performance on predicting multiple ECHO-derived clinical labels.

2. While some questions remain regarding the specifics of the tabular measurements and the model’s zero-shot capabilities, the overall approach is novel and compelling. Leveraging real-world multimodal data to enhance clinical prediction from routinely acquired CT scans has the potential to improve efficiency in clinical workflows and reduce reliance on additional diagnostic modalities.

**Questions To Address In The Rebuttal:**

1. Could the authors elaborate on the motivation for using element-wise summation to fuse modality embeddings in Equation 3, rather than alternative fusion strategies such as concatenation or attention-based fusion? Additional justification or ablation would help clarify this design choice.

2. The image encoder used in X-Cardia is a relatively simple 3D ResNet-50 architecture. Were more expressive or pre-trained CT encoders—such as those from existing CT vision foundation models (e.g., CT-CLIP)—considered or evaluated? If so, it would be informative to report how their performance compares.

3. The manuscript provides limited detail regarding the construction of the tabular embeddings. Could the authors clarify which ECG and ECHO measurements were included, the dimensionality of the resulting tabular embeddings, and whether feature selection or optimization was explored to identify the most informative subsets for model training?

4. While X-Cardia demonstrates strong performance in the 5-shot setting, have the authors evaluated the model in a zero-shot regime? Given the emphasis on representation learning and multimodal alignment, zero-shot performance would be particularly informative for assessing the generalization capability of the learned representations.

5. Do the authors plan to make the dataset publicly available in the future? Given that the dataset appears to be carefully curated and integrates routinely acquired CT imaging with ECG and ECHO measurements, public release could significantly benefit the community and further advance research in multimodal cardiovascular modeling.

---

> ### Author Response · Authors · 2026-01-24
> **Official Comment - 01**
>
> We thank the Reviewer U9PA for these detailed comments and would like to respond to the questions:
>
> ### **Q1: Lack of tabular-only baselines**
>
> **Response:** We thank the reviewer for this thoughtful suggestion. We clarify that the goal of this work is not to predict cardiac labels from ECG or ECHO measurements, which are already strong diagnostic tools, but to enable **opportunistic cardiac screening** from routine, non-gated chest CT acquired for non-cardiac indications. In many clinical settings, ECG and ECHO are unavailable at the time of CT acquisition and are only ordered after suspicion of disease. Accordingly, tabular-only models assume access to information that opportunistic screening seeks to anticipate rather than rely upon.
>
> In **X-Cardia**, ECG and ECHO features are used solely as intermediate supervision during pre-training to shape the CT representation and are not available at inference. We therefore focus our evaluation on CT-only prediction, which reflects the intended deployment scenario and clinical motivation of the method.
>
> ### **Q2: Scalability concerns due to reliance on paired CT–ECG–ECHO data**
>
> **Response:** We agree that access to paired CT–ECG–ECHO data can be a practical constraint. We clarify that such pairing is required only **during the pre-training stage as a one-time supervision cost**, and not at inference, where X-Cardia operates solely on CT, enabling scalable deployment for opportunistic cardiac screening.
>
> Importantly, we now explicitly acknowledge that **retrospectively collected multimodal cohorts may represent a clinically enriched population** and therefore differ in distribution from patients undergoing routine CT alone. This may introduce selection bias and distribution shift during pre-training. We have added a dedicated discussion in the **Discussion and Limitations** section (highlighted in red) and identified bias-aware pre-training, domain adaptation, and validation on more heterogeneous CT-only cohorts as important directions for future work.
>
> ### **Q3: Missing standard deviation**
>
> **Response:** We thank the reviewer for this comment. We now report mean ± standard deviation for all AUROC results across repeated runs in all tables. In addition, we include paired statistical significance testing to quantify the robustness of observed improvements. These results are reported in the main tables and Appendix (highlighted in red).
>
> ### **Q4: Justification for element-wise summation fusion**
>
> **Response:** We clarify that element-wise summation is used to enforce balanced modality contributions while avoiding additional learnable parameters that could bias optimization toward the cleaner tabular signal. We include ablation results comparing summation with concatenation and gated fusion in **Appendix E**, which show comparable or improved stability for summation. This motivation is now explicitly stated in the revised manuscript.
>
> ### **Q5: Use of a simple 3D ResNet-50 encoder**
>
> **Response:** We evaluated a stronger CT vision foundation model by comparing X-Cardia against a **CT-CLIP–pretrained encoder**. As reported in **Figure 5** and discussed in **Section 4.4**, phenotype-guided multimodal pre-training (NW+MMCL) consistently outperforms CT-CLIP across all ten cardiac tasks, with particularly large gains for valvular disease prediction.
>
> We intentionally adopt a standard 3D ResNet-50 backbone to isolate the contribution of phenotype-guided alignment; these results indicate that **cardiac-specific multimodal supervision provides greater gains than encoder complexity alone**. We additionally note (in red) that **future work will explore integrating phenotype-guided alignment with more expressive image encoder architectures** to further improve representation capacity.
>
> ### **Q6: Clarification of tabular embedding construction**
>
> **Response:** We have expanded the manuscript and added an **Appendix H** detailing the tabular embedding construction. Specifically, we list all ECG and ECHO input variables used and report the dimensionality of the learned representations. We also clarify that we did not perform explicit feature selection, as the structured variables are used as intermediate supervision to guide CT representation learning. We note feature subset optimization as future work.

---

> ### Author Response · Authors · 2026-01-24
> **Official Comment - 02**
>
> ### **Q7: Zero-shot evaluation**
>
> **Response:** We appreciate the reviewer’s suggestion and agree that zero-shot evaluation provides an important perspective on representation generalization. Accordingly, we have extended our experiments to include a zero-shot setting and report the results in a **new appendix (Appendix J)**, with corresponding discussion added to the main paper. In this regime, the CT encoder is frozen after multimodal pre-training and evaluated without task-specific fine-tuning. Lower absolute performance in the zero-shot setting reflects the intrinsic difficulty of predicting fine-grained cardiac phenotypes from non-gated chest CT without task-specific supervision, where many functional abnormalities are only indirectly visible. Nevertheless, the consistent improvements over other methods demonstrate that phenotype-guided multimodal alignment enables the model to learn more transferable and task-relevant CT representations.
>
> ### **Q8: Dataset and code availability**
>
> **Response:**  Due to protected health information (PHI) and institutional review board (IRB) constraints, the underlying dataset cannot be made publicly available. To promote reproducibility and enable future research, we are releasing the complete code implementation and pre-trained model weights upon publication.

---

### Author Rebuttal · Authors · 2026-01-24

**Rebuttal:**

In response to reviewers' feedback, we have substantially revised the manuscript. Below, we summarize the key changes in the manuscript. All additions and modifications are highlighted in red in the revised manuscript.

### **Methodology:**

 * U9PA - Q4: Added clarification on why element-wise summation is used for modality fusion, emphasizing balanced contributions and avoidance of optimization bias toward cleaner tabular signals. Referenced new ablation results comparing summation with alternative fusion strategies (Appendix E).

### **Experiments and Results:**

* U9PA - Q3 and Dgqs - Q1: Updated all result tables to report mean ± standard deviation across repeated runs and added paired statistical significance testing (Appendix F).

* U9PA - Q5: Added a comparative analysis against a CT-CLIP pre-trained encoder, reporting results in Figure 4, which show that phenotype-guided multimodal pre-training (NW+MMCL) consistently outperforms CT-CLIP across all ten cardiac tasks, particularly for valvular disease, demonstrating larger gains than encoder scaling alone.

* apw5 - Q1: Clarified that Figure 2 illustrates cross-modal alignment rather than class separation. Added comparative discriminability analyses (Figure 3, Appendix G), including task-specific classifier score distributions and LDA-based projections, showing consistent directional separation between positive and negative cases despite overlap from clinically continuous phenotypes.

* apw5 - Q2: Expanded the interpretability analysis by combining phenotype-conditioned Grad-CAM visualizations.

### **Discussion and Limitations:**

* U9PA - Q2: Explicitly acknowledged that retrospectively collected multimodal cohorts may represent a clinically enriched population, introducing potential selection bias and distribution shift relative to CT-only deployment settings. Added future work directions, including: bias-aware pre-training, domain adaptation strategies, and validation on more heterogeneous cohorts.
* U9PA - Q5: Future extensions to explore more expressive image encoder architectures.

### **Appendix:**

* U9PA - Q7: Added a zero, one, and two-shot evaluation and compared the performances.
* apw5 - Q3: Added details of computational cost and implementation details.
* U9PA - Q6: Added feature specifications.

The GitHub repository has been shared and will be made publicly available upon publication to support reproducibility and future research.

**Supporting Material:**

/attachment/4b21bd06f0d3b5a3db6632d5935785be9fee96aa.pdf

---

### Meta-Review · Area_Chair_MiMd · 2026-02-03

**Recommendation:** Accept (Oral)
**Confidence:** 5

**Metareview:**

This work proposes a multimodal model that integrates routine non-contrast, non-gated computed tomography (CT) images with tabular measurements from echocardiography and electrocardiography, using CLIP-style contrastive pre-training on a cohort of 20,574 patients. At inference time, the model operates using CT data only to predict the presence of cardiac pathologies. The results demonstrate that explicit phenotype alignment improves data-efficient representations and leads to more accurate predictions. This study therefore highlights the potential of routine chest CT for opportunistic cardiac screening.

The discussion and rebuttal phases were handled seriously by the authors, which significantly improved the manuscript overall and led to a consensus among the reviewers regarding the quality of the final version. For these reasons, I have decided to accept this article.

---

### Decision · Program_Chairs · 2026-02-13

Accept (Poster)